# Boosting COVID-19 Image Classification Using MobileNetV3 and Aquila Optimizer Algorithm

**DOI:** 10.3390/e23111383

**Published:** 2021-10-22

**Authors:** Mohamed Abd Elaziz, Abdelghani Dahou, Naser A. Alsaleh, Ammar H. Elsheikh, Amal I. Saba, Mahmoud Ahmadein

**Affiliations:** 1Department of Mathematics, Faculty of Science, Zagazig University, Zagazig 44519, Egypt; 2Artificial Intelligence Research Center (AIRC), Ajman University, Ajman 346, United Arab Emirates; 3Mathematics and Computer Science Department, University of Ahmed DRAIA, Adrar 01000, Algeria; dahou.abdghani@univ-adrar.edu.dz; 4Mechanical Engineering Department, Imam Mohammad Ibn Saud Islamic University, Riyadh 11432, Saudi Arabia; naalsaleh@imamu.edu.sa (N.A.A.); m.ahmadein@f-eng.tanta.edu.eg (M.A.); 5Department of Production Engineering and Mechanical Design, Faculty of Engineering, Tanta University, Tanta 31527, Egypt; 6Department of Histology, Faculty of Medicine, Tanta University, Tanta 31527, Egypt; amal.saba@med.tanta.edu.eg

**Keywords:** feature selection, metaheuristic, atomic orbital search, dynamic opposite-based learning

## Abstract

Currently, the world is still facing a COVID-19 (coronavirus disease 2019) classified as a highly infectious disease due to its rapid spreading. The shortage of X-ray machines may lead to critical situations and delay the diagnosis results, increasing the number of deaths. Therefore, the exploitation of deep learning (DL) and optimization algorithms can be advantageous in early diagnosis and COVID-19 detection. In this paper, we propose a framework for COVID-19 images classification using hybridization of DL and swarm-based algorithms. The MobileNetV3 is used as a backbone feature extraction to learn and extract relevant image representations as a DL model. As a swarm-based algorithm, the Aquila Optimizer (Aqu) is used as a feature selector to reduce the dimensionality of the image representations and improve the classification accuracy using only the most essential selected features. To validate the proposed framework, two datasets with X-ray and CT COVID-19 images are used. The obtained results from the experiments show a good performance of the proposed framework in terms of classification accuracy and dimensionality reduction during the feature extraction and selection phases. The Aqu feature selection algorithm achieves accuracy better than other methods in terms of performance metrics.

## 1. Introduction

In December 2019, COVID-19 was declared as a new coronavirus which resulted in an explosive outbreak in China [1]. Due to its highly contagious characteristics, it swept over more than 220 countries with more than 200 million confirmed cases and more than 4.3 million deaths. This pandemic has the second rank among all documented pandemics based on the number of deaths after the 1918 flu pandemic [2]. More than 40% of these confirmed cases and deaths are reported in only three countries: namely the United States, Brazil, and India, as shown in Figure 1. The symptoms of this disease are fever, dry cough, loss of smell and taste, dyspnea, fatigue, and malaise [3]. It may produce acute complications for persons who suffer from other chronic diseases such as hypertension, respiratory system diseases, autoimmune diseases, diabetes, and cardiovascular diseases.

Diagnosis of COVID-19 infection using X-ray imaging of the chest has been reported as an accurate diagnosis technique [4]. The conventional human-based detecting technique that depends on the technical experience of a physician or radiologist is inefficient, inaccurate, time consuming, limited, and outdated [5]. The implementation of this technique is subjected to human errors, resulting in the misdiagnosing of the disease. This problem is exacerbated in remote regions where there is a lack of expert physicians. The development of advanced artificial intelligence techniques (AI) allows medical researchers and scientists to develop advanced tools, software, and instruments that can help medical radiologists overcome the problems related to human-based detecting techniques [6]. The last two years have seen a surge in the applications of AI in the diagnosis and forecasting of COVID-19 [7,8,9,10,11,12]. Many approaches have been developed to detect and differentiate between COVID-19 disease and conventional viral pneumonia using chest X-ray and CT images [13]. Zhao et al. [14] investigated the relationship between COVID-19 pneumonia and CT images of the chest. The results revealed typical features observed in the examined images of COVID-19 cases; this finding allows researchers to apply AI in the image processing of chest X-rays and CT of COVID-19 cases. Bernheim et al. [15] reported that the CT images of the infected COVID-19 cases are characterized by the existence of typical hallmarks such as consolidative opacities, ground-glass opacities, and crazy-paving patterns. Pezzano et al. [16] developed a convolutional neural network (CNN) to detect ground-glass opacities in the CT images of COVID-19 infected cases. Yasin et al. [17] correlated the disease severity to patients’ sex and age based on X-ray images.

The most common used AI reported in the literature to diagnose COVID-19 infections based on CT or X-ray images is CNN models such as VGG-16, VGG-19, Xception, AlexNet, ResNet50V2, CoroNet, LeNet-5, ResNet18, and ResNet 50 [18,19]. The integration between machine learning methods and the so-called metaheuristic (MH) optimization techniques [20,21] has also been reported in the literature as a correct approach with reasonable computational cost. Canayaz [22] developed a hybrid deep neural network-integrated with metaheuristic optimizers to diagnose COVID-19 infections. A dataset contains three groups of X-ray images, namely normal, pneumonia, and COVID-19, and was used to train the model. The images were preprocessed using the contrast-enhancing technique. Features were extracted using deep learning models, namely GoogleNet, VGG19, AlexNet, and ResNet. The best features were selected using two metaheuristic optimizers, namely grey wolf optimizer and particle swarm optimizer. Then, the features were classified using a support vector machine.

An advanced hybrid classification approach consists of a CNN model and the marine predators optimizer, and the fractional-order algorithm has been developed to detect the infection of COVID-19 based on X-ray images [23]. CNN was used to extract features from the images, while the marine predators optimizer integrated with a fractional-order algorithm was used to select the essential features. The results obtained by the proposed approach were compared with those obtained by other metaheuristic optimizers such as henry gas solubility optimizer, slime mold algorithm optimizer, whale optimization optimizer, particle swarm optimizer, sine cosine algorithm, genetic algorithm, grey wolf optimizer, Harris hawks optimizer, and standalone marine predators optimizer. The proposed approach results revealed its excellent performance compared to the other algorithms in terms of high detection accuracy and low computational cost.

A hybrid metaheuristic optimizing the approach, in which the marine predators optimizer is incorporated with the moth-flame optimizer, was used for image segmentation of COVID-19 cases [24]. The latter optimizer was used as a subroutine in the former optimizer to avoid trapping into local optima. The proposed approach outperformed other advanced optimizers such as the Harris hawks optimizer, grey wolf optimizer, particle swarm optimization, grasshopper algorithm, cuckoo search optimizer, spherical search optimizer, moth-flame optimizer, and standalone marine predators optimizer. An improved cuckoo search optimizer using a fractional calculus algorithm was used to classify X-ray images of COVID-19 cases into normal, COVID-19, and pneumonia patients [25]. Four heavy-tailed distributions, namely Cauchy distribution, Mittag-Leffler distribution, Weibull distribution, and Pareto distribution, were utilized to strengthen the model performance. The proposed model showed its superiority over other feature selection techniques such as the genetic algorithm, Henry gas solubility optimizer, Harris hawks optimizer, salp swarm optimizer, whale algorithm, and grey wolf optimizer.

Another machine learning/metaheuristic optimizing approach was proposed to detect COVID-19 infections based on X-ray images [26]. The important features were extracted from the processed images using a machine learning algorithm called fractional exponent moments. The computational process was accelerated using a multicore computational scheme. Then, a hybrid manta-ray foraging/differential evolution optimizer was used to select the important features. The selection of essential features and excluding irrelevant features help accelerate the classification process, which is accomplished using the k-nearest neighbors technique.

However, most of the presented COVID19 image classification methods have limitations that affect classification accuracy. It has been noticed that these limitations result from either the strategy used to extract the features or the approach used to reduce the number of selected features. Therefore, this motivated us to present an alternative COVID-19 image classification method.

Within this study, we developed an alternative COVID-19 image classification technique that combined the advantages of MobileNetV3 and a new MH technique named Aquila Optimizer (Aqu) [27]. The MobileNetV3 is used to extract the features from the tested images and then, using the binary version of Aquila Optimizer (Aqu) as a feature selection (FS) method, to determine the relevant features. Aqu algorithm has established its performance in several applications, including oil production forecasting [28], and global optimization [29].

The main contributions of this paper are summarized as follows:Develop a COVID-19 cases detection framework by incorporating MobileNetV3 and Aquila Optimizer as feature extraction and selection algorithms, respectively.Propose a new feature selection using the binary version of Aquila Optimizer, in addition, using MobileNetV3 to learn and extract the image embedding from the COVID-19 images.Evaluate the performance of the developed method using two datasets with X-ray and CT images of COVID-19.Compare the efficiency of the developed approach with other methods.

The structure of the remaining parts of this study is as follows: Section 2 introduces the background of MobileNetV3 and Aquila Optimizer algorithm. Section 3 presents the stages of the developed method. The comparison results are given in Section 4. Finally, we introduce the conclusion and future work of the current study in Section 5.

## 2. Background

### 2.1. MobileNetV3

Convolutional neural network architectures have been proposed recently to tackle many different problems and improve their performance in terms of speed and size. Efficient convolutional neural networks implementing the depthwise convolution structure such as NASNet [30], MobileNets [31,32], EfficientNet [33], MnasNet [34], and ShuffleNets [35] are considered as a key technique in many computer vision applications [36,37,38,39] known by fast training process. The depthwise convolutional kernel is a learnable parameter applied to each input channel separately from the training images to extract spatial information. Moreover, depthwise convolutional kernels are shared across all input channels, increasing model efficiency and reducing computation cost. However, the depthwise convolutional kernel size can be difficult to learn, thus increasing the complexity of the training process of the depthwise convolutions. In the upcoming paragraphs, we briefly discuss the recently proposed MobileNetV3 [32] architecture.

Previously developed MobileNetV1 and MobileNetV2 were improved with a new version called MobileNetV3, proposed by Howard et al. [32] using network architecture search (NAS). The used NAS technique called NetAdapt algorithm was used to search for the best kernel size and find the optimized MobileNet architecture to fulfill the low-resourced hardware platforms in terms of size, performance, and latency. The MobileNetV3 introduced several building components and blocks inspired by the previous versions, as shown in Figure 1. In addition, MobileNetV3 possesses a new nonlinearity called hard swish (h−swish), which is a modified version of the sigmoid function introduced in [40]. The h-swish nonlinearity is defined as in Equation (Equation 1), which is employed to minimize the number of training parameters and reduce the model complexity and size.
(1)h−swish(x)=x·σ(x)
(2)$$\sigma \left(x\right)=\frac{ReLU6(x+3)}{6}$$
where σ(x) represents the piece-wise linear hard analog function.

As shown in Figure 1, the MobileNetV3 block contains a core building block called the inverted residual block, which includes a depthwise separable convolution block and a squeeze-and-excitation block [34]. The inverted residual block is inspired from the bottleneck blocks [41], where it uses an inverted residual connection to connect the input and output features on the same channels and improve the features representations with low memory usage. The depthwise separable convolutional contains a depthwise convolutional kernel applied to each channel and a 1×1 pointwise convolutional kernel with batch normalization layer (BN) and the ReLU or h−swish activation functions. The depthwise separable convolutional is used to alter the traditional convolution block and reduce the model capacity. The squeeze-and-excitation (SE) block is used to pay more attention to the relevant features on each channel during training.

### 2.2. Aquila Optimizer (Aqu)

Aquila Optimizer (Aqu) [27] is a new population-based optimizer that is classified as a metaheuristic optimization technique. The mathematical formulation of this optimizer is presented in this section. The social behavior of Aquila inspires the Aqu algorithm during the hunting process of its prey. Like other population-based metaheuristic techniques, Aqu starts with *N* agents with an *X* initial population. This initialization process is executed using the following formula.
(3)$$X_{ij}=r_{1}\times \left(UB_{j}-LB_{j}\right)+LB_{j},\;\;\;i=1,2,.....,N\;j=1,2,\ldots ,Dim$$
where $LB_{j}$ and $UB_{j}$ are the lower and upper bounds of the exploration domain. $r_{1}\in \left[0,1\right]$ is a randomly generated parameter, and Dim is the population size.

Once the population is initialized, the algorithm executes exploitation and exploration processes until the optimal solution is obtained. There are two main implemented strategies during exploitation and exploration processes [27].

The first strategy is implemented to execute the exploration process considering the average agents ($X_{M}$) and the best agent $X_{b}$. This strategy is mathematically formulated as follows:(4)$$X_{i}\left(t+1\right)=X_{b}\left(t\right)\times \left(\frac{1-t}{T}\right)+\left(X_{M}\left(t\right)-X_{b}\left(t\right)*rand\right),$$
(5)$$X_{M}\left(t\right)=\frac{1}{N}\sum_{i=1}^{N}X\left(t\right),\;\forall j=1,2,\ldots ,Dim$$
where *T* is the total number of iterations, while the search process is controlled using $\left(\frac{1-t}{T}\right)$.

In the second strategy, the exploration of the agents is updated based on the Levy flight (Levy(D)) distribution and $X_{b}$. This strategy is mathematically formulated as follows:(6)$$X_{i}\left(t+1\right)=X_{b}\left(t\right)\times Levy\left(D\right)+X_{R}\left(t\right)+\left(y-x\right)*rand,$$
(7)$$Levy\left(D\right)=s\times \frac{u\times \sigma }{{\left|\upsilon \right|}^{\frac{1}{\beta }}},\;\sigma =\left(\frac{\Gamma \left(1+\beta \right)\times sine\left(\frac{\pi \beta }{2}\right)}{\Gamma \left(\frac{1+\beta }{2}\right)\times \beta \times 2^{\left(\frac{\beta -1}{2}\right)}}\right)$$
where β=1.5 and s=0.01, while υ and *u* are randomly generated parameters. In Equation (Equation 6), $X_{R}$ is a randomly selected agent. Moreover, *x* and *y* are used to follow the spiral tracking shape, and they are mathematically formulated as follows:(8)y=r×cos(θ),x=r×sin(θ)
(9)$$r=r_{1}+U\times D_{1},\;\theta =-\omega \times D_{1}+\theta_{1},\;\theta 1=\frac{3\times \pi }{2}$$
where U=0.00565 and ω=0.005. $r_{1}\in \left[0,20\right]$ is a randomly generated parameter.

In [27], the first strategy is utilized to update the agents during the exploitation process based on $X_{M}$ and $X_{b}$, and it is mathematically formulated as follows:(10)$$X_{i}\left(t+1\right)=\left(X_{b}\left(t\right)-X_{M}\left(t\right)\right)\times \alpha -rand+\left(\left(UB-LB\right)\times rand+LB\right)\times \delta ,$$
where δ and α denote the adjustment parameters of exploitation process. rand∈[0,1] is a randomly generated parameter.

In the second strategy, the agent is updated during the exploitation process using the quality function QF, and $X_{b}$, Levy. This strategy is mathematically formulated as follows:(11)$$X_{i}\left(t+1\right)=QF\times X_{b}\left(t\right)-\left(G_{1}\times X\left(t\right)\times rand\right)-G_{2}\times Levy\left(D\right)+rand\times G_{1},$$
(12)$$QF\left(t\right)=t^{\frac{2\times rand()-1}{{\left(1-T\right)}^{2}}}$$

Furthermore, $G_{1}$ specifies the employed motions during tracking the best solution, and it is given as:(13)$$G_{1}=2\times rand\left(\right)-1,\;G_{2}=2\times \left(1-\frac{t}{T}\right)$$
rand is a function that generates random values, and $G_{2}$ specifies decreased values from 2 to 0, and it is given as:(14)$$G_{2}=2\times \left(1-\frac{t}{T}\right)$$

## 3. Proposed Framework

In this section, the general framework of the developed COVID-19 image classification method is described.

### 3.1. MobileNetV3 for Feature Extraction

The fine-tuning process of MobileNetV3 and feature extraction phase are described in this section. The main objective is to extract relevant image embeddings relying on a pretrained model on different COVID-19 image datasets. Meanwhile, the extracted image embedding in this phase is fed into the feature selection phase, discussed in the next section. Compared to previous studies, the feature selection phase employs a new swarm optimization technique to enhance the recognition accuracy, select only essential features, and reduce the features representation space of the overall proposed framework.

As described in Section 2.1, efficient convolutional neural networks such as MobileNetV3 [32] act as suitable models to perform image recognition where they can act as a core component in the feature extraction phase. We used a pretrained model of MobileNetV3 trained on the ImageNet dataset to avoid training the model from scratch and speed up the learning process. More specifically, the MobileNetV3-Large pretrained model was used in our experiments and adapted to the COVID-19 recognition task via transfer learning and fine-tuning. We follow the standard procedure to fine-tune the MobileNetV3 model and extract the relevant image embeddings. First, we change the top two output layers of the MobileNetV3 model used for image classification with a 1×1 point-wise convolution to extract images features. The 1×1 point-wise convolution can act as a multilayer perceptron (MLP) to perform image classification or dimensionality reduction by integrating different nonlinearity operations. In addition, other 1×1 point-wise convolutions have been added on the top of the model for fine-tuning the model’s weights on different datasets based on the classification task. Second, after fine-tuning the model, we flatten the output of the 1×1 point-wise convolution used for feature extraction to generate image embeddings with the size of 128 for each image in the dataset. Lastly, the extracted image embeddings are fed into the feature selection phase.

Figure 2 shows the architecture of the modified MobileNetV3 for COVID-19 images feature extraction. The feature extraction phase was performed after fine-tuning the model for 100 epochs during ten randomly initialized runs where we used the model resulting in the highest classification accuracy on each dataset. A batch of size 32 and a stochastic gradient descent approach named RMSprop were used to fine-tune the model with a learning rate set to $1\times 10^{-4}$. Data augmentation was employed during the data preprocessing phase to overcome overfitting and improve the model’s generalization. The data augmentation transformation such as random crop, random horizontal flip, color jitter, and random vertical flip was used alongside original image resizing to shape 224×224.

### 3.2. Developed Aqu FS Algorithm

To apply the Aqu algorithm as an FS method, its binary version is developed as given in Figure 3. The main target of this conversion is to prepare the Aqu algorithm for working with the discrete problem since its original version is implemented to work with real-valued problems only. There are two stages of Aqu as FS technique, and the details of each stage are given in the following sections.

#### 3.2.1. First Stage: Learning of Model

This stage aims to use the training set to learn the model to select the most relevant features, and in this study, we used 70% from the dataset as a training set. The first process in this stage is to set the initial value for the population *X*, which contains *N* agents. This process is defined as:(15)$$X_{i}=rand*\left(U-L\right)+L,\;i=1,2,\ldots ,N,\;j=1,2,\ldots ,N_{F}$$
where $N_{F}$ represents the number of features, whereas *U* and *L* are the limits of search domain.

The next step is to obtain the binary form for each $X_{i}$, and this was produced using Equation (Equation 16).
(16)$$BX_{ij}=\left\{\begin{array}{cc} 1 & if\;X_{ij}>0.5\\ 0 & otherwise \end{array}\right.$$
Then, the fitness value $Fit_{i}$ of each $X_{i}$ is evaluated using the following formula:(17)$$Fit_{i}=\lambda \times \gamma_{i}+\left(1-\lambda \right)\times \left(\frac{|BX_{i}|}{N_{F}}\right),$$
In Equation (Equation 17), $|BX_{i}|$ stands for the number of features (i.e., the ones in $BX_{i}$), whereas $\gamma_{i}$ represents the classification error using the KNN classifier that used the reduced training set based on $BX_{i}$. In addition, λ is a weight value used to balance between the two objectives in Equation (Equation 17) (i.e., minimizing the selecting features and reducing the error of classification).

After that, the agent with the best fitness value ($Fit_{b}$) is considered the best agent $X_{b}$. The $X_{b}$ agent is used to update the other agents according to the operators of the Aqu algorithm as discussed in Equations (Equation 4)–(Equation 14).

The next step is to check if the terminal conditions are met, then $X_{b}$ is returned; otherwise, updating the solutions is conducted again.

#### 3.2.2. Second Stage: Evaluation of the Selected Features

Within this stage, the relevant features in the best solution $X_{b}$ are used to reduce the testing set used as input to the KNN classifier. Later, we compute the performance of the predicted output using various performance measures.

## 4. Experimental Results

### 4.1. Dataset Description

This section describes the datasets used in the COVID-19 detection task and the distribution of their corresponding samples. The datasets include two types of images: X-ray and CT scan images (computed tomography scan), where Figure 4 shows examples from each dataset. Our experiments used three different datasets to train and fine-tune the feature extraction model, namely the COVID-CT dataset (dataset1), the COVID-XRay-6432 dataset (dataset2), and the COVID-19 radiography dataset (dataset3). We keep the same data split after extracting image embeddings from each dataset which are fed to a feature selection and classification phase. In the following section, a detailed description of each dataset is given.
COVID-CT dataset: This dataset was collected from two sources, including research papers (for training) and original CT scans donated by hospitals (for testing). For the research papers, the authors [42] collected 760 preprints from two databases including medRxiv https://www.medrxiv.org/ (accessed on 12 October 2021) and bioRxiv https://www.biorxiv.org/ (accessed on 12 October 2021). The preprints were collected from papers posted from 19 January to 25 March 2020. In total, 349 CT images labeled as positive were collected from 216 patient cases for COVID-19. In addition, the authors collected 397 negative CT images (Non-Covid19) to build their dataset for a binary classification task from sources including MedPix https://medpix.nlm.nih.gov/home (accessed on 12 October 2021) database, the LUNA7 https://luna16.grand-challenge.org/ (accessed on 12 October 2021) dataset, the Radiopaedia https://radiopaedia.org/articles/covid-19-3 (accessed on 12 October 2021) website, and PubMed https://www.ncbi.nlm.nih.gov/pmc/ (accessed on 12 October 2021) Central (PMC). Table 1 lists the number of positive and negative Covid-19 CT images used in our experiments.COVID-XRay-6432 dataset: The dataset is publicly available on Kaggle https://www.kaggle.com/prashant268/chest-xray-covid19-pneumonia (accessed on 12 October 2021) and was gathered from various public resources. The dataset includes 6432 X-ray COVID-19 images distributed on three classes which are COVID-19, PNEUMONIA, and NORMAL (Non-COVID). The training set comprises 80% of the dataset, and the test set comprises 20% of the dataset. In our experiments, 15% of the training sample is used in the validation set and fine-tuning. Table 2 lists the number of samples in each class.COVID-19 radiography dataset: The dataset was collected by a team of researchers from different countries and universities, including Qatar, Bangladesh, Pakistan, and Malaysia, collaborating with medical doctors. The dataset is freely available and frequently updated on Kaggle https://www.kaggle.com/tawsifurrahman/covid19-radiography-database (accessed on 12 October 2021). The dataset consists of 21,165 chest X-ray (CXR) COVID-19 images distributed on four categories which are COVID19, lung opacity, viral pneumonia, and NORMAL (Non-COVID). In our experiment, we randomly split the data into 70%, 10%, and 20% for training, validation, and testing sets, respectively. Table 3 lists the number of samples in each class after splitting the data.

### 4.2. Performance Metrics

To assess the accuracy of the developed model, some statistical parameters were computed, such as the mean of best values, the mean of the worst values (Max), standard deviation, and computational time elapsed during the selection of features. Then, statistical measures were computed during the classification phase. The mathematical form of these measures are given as:(18)$$\begin{array}{cc} & Accuracy=\frac{\mathrm{TP}+\mathrm{TN}}{\mathrm{TP}+\mathrm{TN}+\mathrm{FP}+\mathrm{FN}}\\ \; & Sensitivity=\frac{\mathrm{TP}}{\mathrm{TP}+\mathrm{FN}}\\ \; & Specificity=\frac{\mathrm{TN}}{\mathrm{TN}+\mathrm{FP}}\\ \; & F_{Score}=2\times \frac{\mathrm{Specificity}\times \mathrm{Sensitivity}}{\mathrm{Specificity}+\mathrm{Sensitivity}} \end{array}$$
where “TP” is the abbreviation of true positives and represents the positive COVID-19 images labeled using the proposed classifier correctly. “TN” stands for the true negative samples and represents the negative COVID-19 images that were labeled using the proposed classifier correctly. “FP” is the abbreviation of false positives and represents the positive COVID-19 images labeled using the proposed classifier incorrectly, while “FN” is the abbreviation of false negatives and represents the negative COVID-19 images that were labeled using the proposed classifier incorrectly.
Best accuracy:
(19)$$Best_{acc}=\max_{1\leq i\leq r}Accuracy$$
where *r* denotes the run numbers. $Fit_{i}$ represents a fitness function value.

To validate the performance of Aqu as an FS method, its results were compared with other well-known FS methods based on MH techniques. For example, whale optimization algorithm (WOA) [43], moth-flame optimization (MFO) [44,45], firefly algorithm (FFA) [46], bat algorithm (BAT) [47], hunger games search (HGS) [48], transient search optimization (TSO) [49], and Aquila Optimizer (Aqu) [27]. In this paper, the parameters of these FS methods are assigned based on the original implementation of each method. However, the common parameters, such as the number of iterations and population size, are set to 20 and 15, respectively. In addition, each FS method conducted 25 runs for a fair comparison between them. All DL training and feature extraction phases were conducted on a GPU (Graphics processing unit) of type GTX1080 from Nvidia, while the feature selection phase has experimented on the Google collaboratory platform. For a proper validation of the framework, other DL models such as DenseNet, VGG19, and EfficientNet were exploited as backbone feature extraction methods using their standard architecture and parameters.

### 4.3. Results and Discussion

In this subsection, the performance of the developed model is evaluated using two datasets as given in Table 4 and Table 5 and Figure 5 and Figure 6. In general, it can be noticed from Table 4 that the developed method can improve the performance of classification accuracy among the two tested datasets. For example, to analyze the performance of the developed Aqu over Dataset1, the following points can be observed: firstly, the accuracy of Aqu is better than other methods, which nearly have a difference between the best second algorithm (i.e., HGS) with 1.2%. Secondly, Aqu has a higher Recall, Precision, and F1 score overall than the comparative FS methods such as HGS and BAT, which allocated the second and third ranks, respectively. Similar to dataset1, the efficiency of Aqu using dataset2 is the best in terms of classification accuracy, followed by BAT and HGS. In addition, the recall value of HGS and MFO is better than all other methods (i.e., WOA, FFA, BAT, and TSO) except for the developed Aqu method, whereas the precision value of WOA and TSO allocate the second rank after the Aqu method, which allocates the first rank. In order to analyze the results of the F1-score obtained using the second dataset, it can be observed that the HGS is better than other algorithms, which allocate the second rank after the Aqu algorithm.

Figure 5 depicts the average of each algorithm in terms of accuracy, recall, precision, and F1-score. It can be seen from this figure that the average of the Aqu algorithm over the two datasets is better than other methods in terms of performance measures.

Moreover, the time computational of the FS methods is computed to justify their time complexity as given in Table 5. We can notice from the CPU time(s) values that the developed Aqu algorithm has the shortest time in dataset1. However, the CPU time(s) of the Aqu algorithm over datasets is the second-best one that followed the TSO algorithm. Meanwhile, the efficiency of Aqu to reduce the number of features is observed from the number of selected features (i.e., the #FS column). Aqu has the smallest number of features, 130 and 140 at Dataset1 and Dataset2, respectively. In addition, from Figure 6 which shows the average over the two datasets in terms of CPU time(s) and #FS, the high superiority of Aqu over other methods can be seen.

### 4.4. Comparison with Other CNN Types

In this section, the performance of the developed method that combines the MobileNetV3 and Aqu is compared with the other three CNN types networks. These network include VGG19 [50] (Visual Geometry Group), DenseNet [51], and EfficientNet [33]. The main aim of this comparison is to assess the ability of MobileNetV3 to extract the relevant features.

The comparison results between the MobileNetV3 and other CNN types are given in Table 6. From these results, it can be seen that the MobileNetV3 can provide better performance than other CNNs followed by DenseNet that has a high ability to extract relevant features better than the other two networks (i.e., VGG and EfficientNet). The same observation can be noticed in Figure 7, which depicts the average of the accuracy in all the tested FS methods using the feature extracted from each CNN type. In addition, the performance of Aqu based on MobileNetV3 in terms of accuracy among the two tested datasets is given in Figure 8. From these averages, it can be noticed that the developed method provides better results than others. In addition, the ability of Aqu to increase the accuracy classification is better than other FS methods when using different CNN types.

### 4.5. Influence of the Size of COVID19 Dataset

In this section, the influence of using a large number of images on the performance of the developed method is assessed using a third dataset (i.e., COVID-19 radiography) described in Section 4.1.

Table 7 shows the average of the results in terms of performance measures for each FS algorithm that depends on the features extracted using MobileNetV3. From these results, one can reach the following observations: firstly, Aqu has a high ability to enhance the classification accuracy; in addition, it can reduce the number of features required to increase the classification accuracy. However, the Aqu allocates the second rank after TSO in CPU time (s) required to determine the relevant features.

## 5. Conclusions

This study developed a framework to detect the COVID-19 cases from X-ray and CT images using three datasets with a considerable amount of samples. The proposed framework depends on the combination of the MobileNetV3 DL model and metaheuristic (MH) techniques. Furthermore, three other DL networks were included in our experiments, namely VGG19, DenseNet, and EfficientNet. For instance, MobileNetV3 was used to extract the features from all existing images in each dataset. By contrast, a new MH technique named Aquila Optimizer (Aqu) was proposed for feature selection (FS) by converting it to binary. The extracted image embeddings from each DL network were fed to the FS algorithm for feature space reduction and classification performance improvement. To justify the performance of the developed method, three datasets are used with different characteristics since they represent X-ray and CT COVID-19 images collected from different sources. The comparison results illustrated the high performance of the developed method based on the Aqu method over the other competitive methods.

Besides the promising results, the developed method can be extended to other applications such as agriculture, remote sensing, galaxy classification, and other image classification tasks.

## Figures and Tables

**Figure 1 entropy-23-01383-f001:**
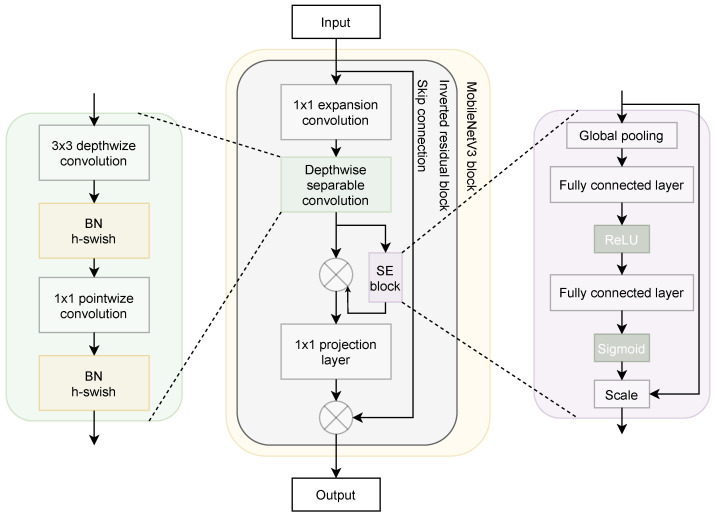
The structure of MobileNetV3 blocks and components.

**Figure 2 entropy-23-01383-f002:**
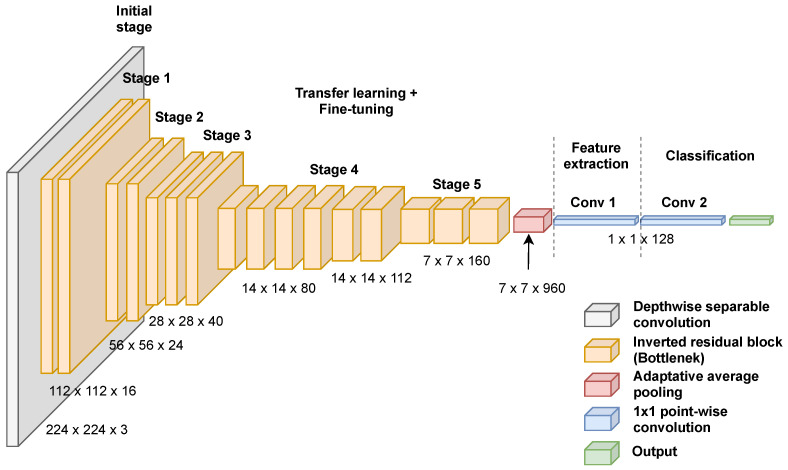
The architecture of MobileNetV3 used for feature extraction.

**Figure 3 entropy-23-01383-f003:**
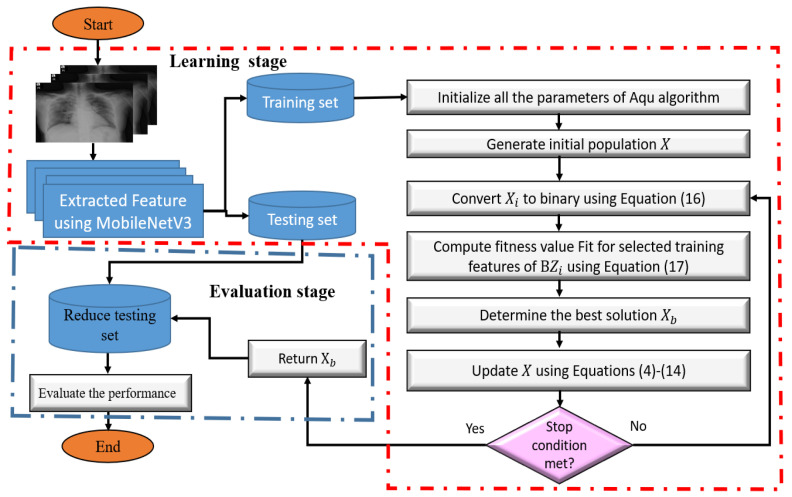
Steps of Aqu for FS problem.

**Figure 4 entropy-23-01383-f004:**
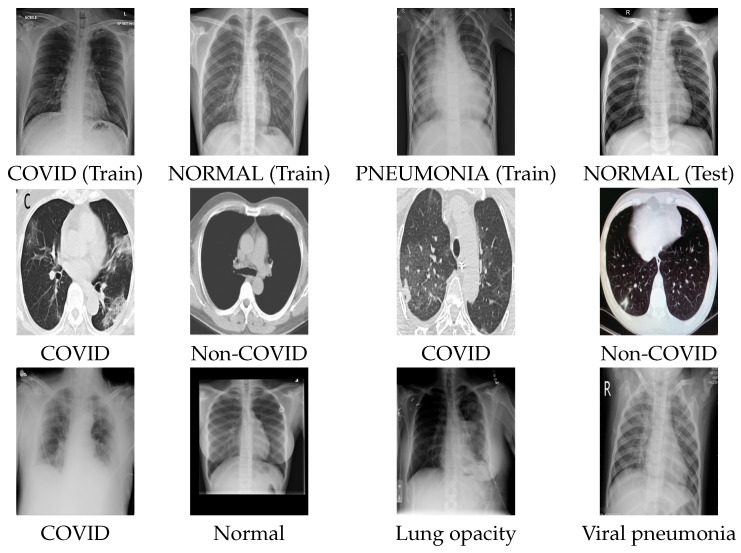
**1st row**: COVID-XRay-6432 dataset samples, **2nd row**: COVID-CT dataset samples, and **3rd row**: COVID-19 radiography dataset samples.

**Figure 5 entropy-23-01383-f005:**
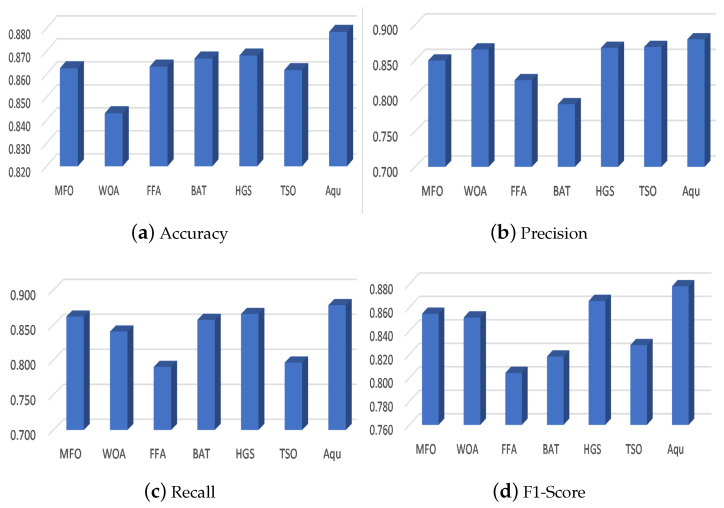
Average of the competitive algorithms in terms of (**a**) Accuracy, (**b**) Precision (**c**) Recall, and (**d**) F1-score.

**Figure 6 entropy-23-01383-f006:**
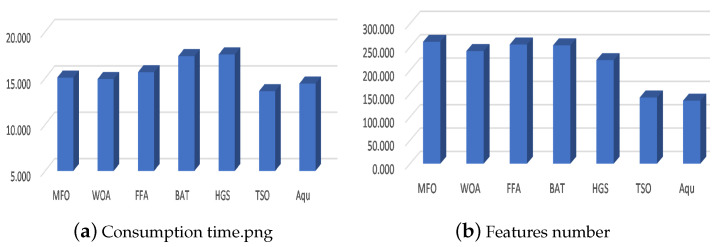
Average of each algorithm among two datasets in terms of (**a**) CPU time(s) and (**b**) number of selected features.

**Figure 7 entropy-23-01383-f007:**
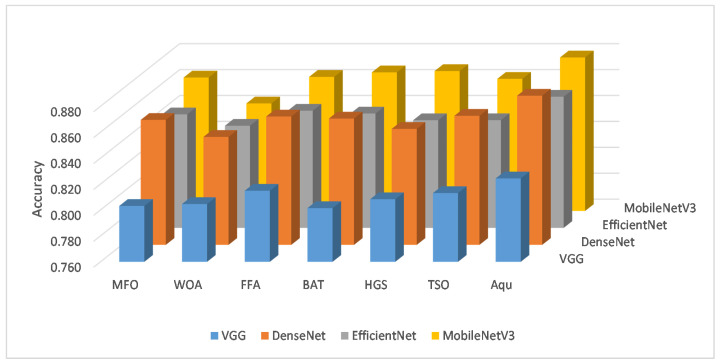
Average of each CNN type overall the FS methods.

**Figure 8 entropy-23-01383-f008:**
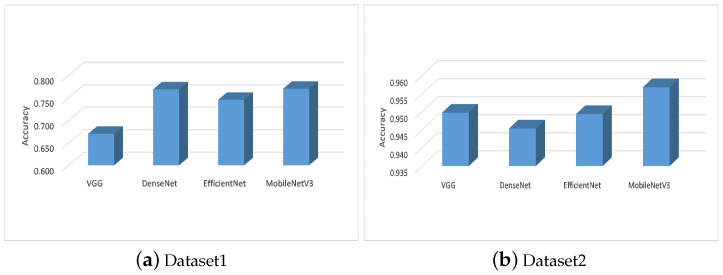
Average of each CNN type overall the FS methods.

**Table 1 entropy-23-01383-t001:** COVID-CT dataset samples distribution.

	Class	Train	Validation	Test
# patients	COVID	130	32	54
	Non-COVID	105	24	42
# images	COVID	191	60	98
	Non-COVID	234	58	105

**Table 2 entropy-23-01383-t002:** COVID-XRay-6432 dataset samples distribution.

	Class	Train	Test
# images	COVID	460	116
	Non-COVID	1266	317
	PNEUMONIA	3418	855

**Table 3 entropy-23-01383-t003:** COVID-19 radiography dataset samples distribution.

	Train	Validation	Test
# images	15,238	1694	4233

**Table 4 entropy-23-01383-t004:** Comparison between Aqu and other methods in terms of accuracy, recall, precision, and Fs-score.

	Dataset1				Dataset2			
	Accuracy	Recall	Precision	F1-Score	Accuracy	Recall	Precision	F1-Score
MFO	0.769	0.769	0.771	0.767	0.957	0.956	0.928	0.942
WOA	0.761	0.761	0.764	0.760	0.925	0.920	0.967	0.943
FFA	0.769	0.769	0.771	0.767	0.958	0.812	0.873	0.841
BAT	0.771	0.771	0.775	0.769	0.963	0.944	0.802	0.867
HGS	0.773	0.773	0.776	0.772	0.963	0.958	0.959	0.959
TSO	0.766	0.766	0.770	0.764	0.958	0.827	0.967	0.892
Aqu	0.783	0.783	0.785	0.782	0.974	0.974	0.974	0.974

**Table 5 entropy-23-01383-t005:** Comparison between Aqu and other methods in terms of CPU time(s) and number of selected features.

	Dataset1		Dataset2	
	CPU Time (s)	#FS	CPU Time (s)	#FS
MFO	3.481	278.5	26.654	243.5
WOA	3.260	248.5	26.572	234
FFA	4.294	260.5	27.009	250
BAT	3.602	256.5	31.179	250.5
HGS	4.115	281	31.028	162.5
TSO	3.197	141.5	24.013	142
Aqu	3.123	130	25.737	140

**Table 6 entropy-23-01383-t006:** Comparison with other CNN types.

	Dataset1				Dataset2			
	VGG	DenseNet	EfficientNet	MobileNetV3	VGG	DenseNet	EfficientNet	MobileNetV3
MFO	0.667	0.766	0.757	0.769	0.939	0.947	0.938	0.957
WOA	0.672	0.751	0.742	0.761	0.937	0.936	0.936	0.925
FFA	0.670	0.784	0.742	0.769	0.960	0.934	0.959	0.958
BAT	0.667	0.756	0.761	0.771	0.935	0.959	0.935	0.963
HGS	0.672	0.764	0.742	0.773	0.944	0.935	0.945	0.963
TSO	0.665	0.785	0.725	0.766	0.961	0.934	0.961	0.958
Aqu	0.676	0.777	0.751	0.783	0.972	0.973	0.971	0.974

**Table 7 entropy-23-01383-t007:** Performance of FS methods using COVID-19 radiography dataset.

	Accuracy	Recall	Precision	F1-Score	CPU Time (s)	#FS
MFO	0.889	0.897	0.840	0.868	15.347	61
WOA	0.886	0.885	0.828	0.855	15.593	58.5
FFA	0.910	0.885	0.828	0.855	15.632	56
BAT	0.887	0.909	0.852	0.880	18.185	57
HGS	0.894	0.884	0.827	0.855	18.306	69
TSO	0.910	0.884	0.827	0.855	15.179	63
Aqu	0.924	0.924	0.866	0.894	15.290	57

## Data Availability

The data can be obtained from https://www.kaggle.com/prashant268/chest-xray-covid19-pneumonia, https://www.medrxiv.org/, https://www.kaggle.com/tawsifurrahman/covid19-radiography-database (accessed on 12 October 2021).

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
