# Peer review of "Boosting COVID-19 Image Classification Using MobileNetV3 and Aquila Optimizer Algorithm"

_entropy, 2021, doi:10.3390/e23111383_

Round 1
Reviewer 1 Report
The manuscript titled "Boosting COVID-19 Image Classification using MobileNetV3 and Aquila Optimizer Algorithm" talks about using a new optimizer algorithm for COVID-19 image classification and tested it using two datasets X-ray and CT imaging. The manuscript deals with a very significant and a relevant problem during the ongoing pandemic. However some major changes are required before the manuscript can be accepted for publication.
- The introduction is too long and reads like a review paper. Please reduce the length of it and be concise in the information you want to convey.
2. Be clear on the motivation of your work. Compared to the various other machine learning models what additional value is your model is adding?
3. What is the difference between the two datasets? The performance of dataset 2 is much higher than dataset 1. Why is the difference?
4. The authors have compared the Aqu with various other parameters. However, I suggest the authors to compare their model with commonly used model such as AlexNet, VGG-19, VGG-16, Resnet etc.
5. I also highly suggest the authors to use the datasets with more images.
6. The discussion section needs to be written better.
Author Response
The manuscript titled "Boosting COVID-19 Image Classification using MobileNetV3 and Aquila Optimizer Algorithm" talks about using a new optimizer algorithm for COVID-19 image classification and tested it using two datasets X-ray and CT imaging. The manuscript deals with a very significant and a relevant problem during the ongoing pandemic. However some major changes are required before the manuscript can be accepted for publication.
- The introduction is too long and reads like a review paper. Please reduce the length of it and be concise in the information you want to convey.
Response
Thanks for this valuable recommendation and we followed it by reducing the introduction
- Be clear on the motivation of your work. Compared to the various other machine learning models what additional value is your model is adding?
Response
We improved the motivation and compared the proposed method with other methods as in section 4.4.
- What is the difference between the two datasets? The performance of dataset 2 is much higher than dataset 1. Why is the difference? DAHOU
Response
Thank you for your comments. The dataset are different in the type of collected image. In dataset1 images are CT image whereas in dataset2 the collected images are X-ray image. The detailed differences between the two types are illustrated in Figure 4.1.
- The authors have compared the Aqu with various other parameters. However, I suggest the authors to compare their model with commonly used model such as AlexNet, VGG-19, VGG-16, Resnet etc. DAHOU
Response
Thank you for your comments. More experiments have been included in the revised version where we conduct more experiments using DL models such as VGG19, DenseNet, and EfficientNet.
- I also highly suggest the authors to use the datasets with more images.
Response
Thanks for this valuable suggestion and we added section 4.5 to study the influence of dataset with more images on the performance of developed method.
- The discussion section needs to be written better.
Response
Thank you for your comment and we improved the discussion in the new version of the manuscript.

Reviewer 2 Report
This research focus on the COVID-19 has its urgency, and at the same time, it is very close to the needs of the public and the medical community. But, AI has been used in medical imaging for many years, and the use of new algorithms to boosting COVID-19 image classification lacks innovation. Anyway, based on the authors' rigorous and good presentation of the research process, clear background statement, and solid literature reviews, etc. I believe this result can make a concrete contribution in this area. The comparison results of this paper illustrated the high performance of the developed method based on Aqu method over the other competitive methods. However, here are some suggestions, hope to help the quality of the paper, be more friendly and specific to the readers.
- The pictures and tables in this paper are well presented, clear and easy to read.
- There is not much discussion on the analysis results in 4.3, and increasing the discussion content can improve the contribution of the paper. For example, it is suggested that the type of algorithm applied to the features is explained based on the research results.
- It is recommended that the authors explain which hardware equipment is used for testing/training to enhance the comparison results in terms of the link to performance.
Author Response
- The pictures and tables in this paper are well presented, clear and easy to read.
Response
Thanks for this positive comment.
- There is not much discussion on the analysis results in 4.3, and increasing the discussion content can improve the contribution of the paper. For example, it is suggested that the type of algorithm applied to the features is explained based on the research results.
Response
Thank you for your comment and we improved the discussion in the new version of the manuscript.
- It is recommended that the authors explain which hardware equipment is used for testing/training to enhance the comparison results in terms of the link to performance. DAHOU
Response
Thank you for your comments. The requested information have been added to the revised manuscript under the performance measure section.

Round 2
Reviewer 1 Report
The authors have addressed my comments sufficiently. I suggest them to do one more round of proof reading as there are minor grammatical errors. Otherwise, I recommend the manuscript to be accepted for publication.
Author Response
Reviewer 1
The authors have addressed my comments sufficiently. I suggest them to do one more round of proof reading as there are minor grammatical errors. Otherwise, I recommend the manuscript to be accepted for publication.
Response
Thanks for your valuable comment and we fixed the grammatical errors.

This manuscript is a resubmission of an earlier submission. The following is a list of the peer review reports and author responses from that submission.